# TXNIP Promotes Human NK Cell Development but Is Dispensable for NK Cell Functionality

**DOI:** 10.3390/ijms231911345

**Published:** 2022-09-26

**Authors:** Eva Persyn, Sigrid Wahlen, Laura Kiekens, Sylvie Taveirne, Wouter Van Loocke, Els Van Ammel, Filip Van Nieuwerburgh, Tom Taghon, Bart Vandekerckhove, Pieter Van Vlierberghe, Georges Leclercq

**Affiliations:** 1Laboratory of Experimental Immunology, Department of Diagnostic Sciences, Ghent University, 9000 Ghent, Belgium; 2Cancer Research Institute Ghent (CRIG), 9000 Ghent, Belgium; 3Department of Biomolecular Medicine, Ghent University, 9000 Ghent, Belgium; 4Department of Pharmaceutics, Ghent University, 9000 Ghent, Belgium

**Keywords:** human natural killer (NK) cells, TXNIP, NK cell development

## Abstract

The ability of natural killer (NK) cells to kill tumor cells without prior sensitization makes them a rising player in immunotherapy. Increased understanding of the development and functioning of NK cells will improve their clinical utilization. As opposed to murine NK cell development, human NK cell development is still less understood. Here, we studied the role of thioredoxin-interacting protein (TXNIP) in human NK cell differentiation by stable TXNIP knockdown or overexpression in cord blood hematopoietic stem cells, followed by in vitro NK cell differentiation. TXNIP overexpression only had marginal effects, indicating that endogenous TXNIP levels are sufficient in this process. TXNIP knockdown, however, reduced proliferation of early differentiation stages and greatly decreased NK cell numbers. Transcriptome analysis and experimental confirmation showed that reduced protein synthesis upon TXNIP knockdown likely caused this low proliferation. Contrary to its profound effects on the early differentiation stages, TXNIP knockdown led to limited alterations in NK cell phenotype, and it had no effect on NK cell cytotoxicity or cytokine production. Thus, TXNIP promotes human NK cell differentiation by affecting protein synthesis and proliferation of early NK cell differentiation stages, but it is redundant for functional NK cell maturation.

## 1. Introduction

Since their discovery in 1975, natural killer (NK) cells have proven to play an important role in the first line defense against virally infected and transformed cells [1]. NK cell recognition of aberrant cells is mediated by an array of germline encoded receptors that transmit either activating or inhibiting signals controlling NK cell effector functions. The activation of NK cells results in the release of cytotoxic granules containing perforin and granzymes, leading to direct lysis of target cells. NK cells also have an immunomodulating function as they are potent producers of inflammatory cytokines, such as interferon (IFN)-γ and tumor necrosis factor (TNF)-α [1,2,3]. NK cells constitute 5–15% of peripheral blood lymphocytes, and like all leukocyte populations, NK cells derive from self-renewing pluripotent hematopoietic stem cells (HSC) that reside in the bone marrow. It was first hypothesized that, as in mice, maturation of human NK cells also occurs exclusively in the bone marrow. However, evidence suggests that CD34^+^CD45RA^+^ hematopoietic progenitor cells leave the bone marrow to further develop in secondary lymphoid tissue, such as lymph nodes and tonsils [4,5]. Distinct developmental stages towards mature NK cells have been described based on the expression of CD34, CD117, CD94, CD56, and CD16 [6]. Stage 1 (CD34^+^CD45RA^+^CD117^−^CD94^−^) and stage 2 (CD34^+^CD45RA^+^CD117^+^CD94^−^) cells are multipotent as they have T cell, dendritic cell, and NK cell developmental potential. Expression of the IL-2 receptor β chain (CD122) by stage 3 cells (CD34^−^CD117^+^CD94^−^) makes them responsive to IL-15 and marks a commitment to the NK cell lineage [7,8]. Acquisition of CD94 indicates the transition to mature NK cells, i.e., stage 4 (CD34^−^CD56^bright^CD94^+^CD16^−^) and stage 5 (CD34^−^CD56^dim^CD94^+^CD16^+^) cells. Stages 4 and 5 are considered functionally mature as they are cytotoxic and able to produce cytokines. 

Commitment, development, maturation, and function of NK cells is regulated by a complex network of several proteins, including transcription factors, cytokines, and others. Intensive research using genetic manipulated mice has identified several essential factors in murine NK cell development [9,10,11]. There are, however, important differences between human and murine NK cell development, leaving knowledge on regulation of human NK cell development still limited. The innate ability of NK cells to kill cancer cells makes them an emerging tool in immunotherapy [12]. Gaining a more complete understanding of the factors that dictate NK cell development and function can be important to develop effective NK cell-based therapies. 

Vitamin D3 upregulated protein 1 (VDUP1) was originally discovered as a protein upregulated in the human leukemia cell line HL-60 by vitamin D3 treatment [13]. Later, the protein was identified as a thioredoxin binding partner and renamed thioredoxin binding protein 2 (TBP-2) or thioredoxin-interacting protein (TXNIP). It belongs to the family of α-arrestin proteins, and it is ubiquitously expressed. As its name implies, TXNIP directly interacts with the redox active domain of thioredoxin, thereby inhibiting its disulfide reducing activity [14]. In HSC, TXNIP has unexpectedly been reported to exert important antioxidant effects [15]. However, redox-independent, tissue-specific TXNIP functions have also been widely observed. TXNIP inhibits cellular glucose uptake by regulating expression and localization of the glucose transporter Glut1 [16]. In tumorigenesis, TXNIP is seen as a negative regulator, as its expression is significantly downregulated in various tumors, such as breast, renal, and gastrointestinal cancers [14]. Regarding murine NK cell development, Lee et al. showed that TXNIP is required in this process as *Txnip* knockout mice have profoundly reduced NK cell numbers in the bone marrow as well as in the spleen and lung, while T and B cell numbers are not affected. The NK cells that do develop in *Txnip* knockout mice show greatly decreased Ly49 and CD122 expression and also have reduced cytotoxic activity [17]. Currently, the role of TXNIP in human NK cell biology remains unknown.

Here, we analyzed the role of TXNIP in human NK cell development using in vitro HSC-based differentiation cultures, in which the starting HSC were stably transduced with either TXNIP knockdown or TXNIP overexpression vectors. We show that decreased TXNIP expression strongly reduces human NK cell differentiation. Transcriptome analysis indicates that TXNIP knockdown reduces both cell proliferation and protein synthesis, which is experimentally confirmed. The remaining NK cells in the TXNIP knockdown cultures display limited phenotypic changes and have normal functional capacities, including cytotoxicity and cytokine production. Thus, TXNIP promotes human NK cell development, but is dispensable for NK cell functionality.

## 2. Results

### 2.1. TXNIP Promotes Human NK Cell Development

To explore the role of TXNIP in human NK cell development, we first examined basal TXNIP expression levels in the successive NK cell developmental stages. Using RT-qPCR, the gene expression of TXNIP in human cord blood-derived HSC and in in vitro-generated NK cell stages 1 to 5 was determined (Figure 1A). TXNIP was highly expressed in HSC, showed decreased expression levels in stage 1 and stage 2 cells, but was again upregulated in stage 3 to stage 5 cells. 

To elucidate the potential role of TXNIP in human NK cells, we assessed the effects of TXNIP knockdown and TXNIP overexpression in cord blood-derived HSC on their subsequent development into NK cells. For the loss-of-function experiments, HSC were transduced with a lentiviral vector encoding enhanced green fluorescent protein (eGFP) containing either TXNIP-specific shRNA or a non-targeting control shRNA, while in the gain-of-function experiments, HSC were transduced with a retroviral vector containing TXNIP cDNA or the empty IRES-eGFP control vector. Successfully transduced HSC (CD34^+^lineage^−^(CD3/CD14/CD19/CD56)CD45RA^−^eGFP^+^) were sorted after 48 h (day 0 of culture) and induced to stepwise differentiate in vitro into NK cells using the well-established EL08-1D2 coculture system [18]. Transduction of HSC with TXNIP shRNA led to significant reduction in TXNIP mRNA expression in HSC, stage 3 cells, and mature NK cells. Transduction of HSC with TXNIP cDNA showed a strong increase in TXNIP mRNA in HSC and stage 3 cells, but no difference was observed in NK cells (Figure 1B). However, Western blot analysis of sorted NK cells from TXNIP overexpression cultures showed that protein levels were significantly increased compared to the control condition. As expected, TXNIP knockdown significantly reduced TXNIP protein expression in NK cells (Figure 1C).

Absolute cell numbers of the different developmental stages were determined at weekly timepoints of the culture (Figure 1D). TXNIP knockdown led to a significant reduction in the absolute cell numbers of HSC, stage 1, stage 2, and stage 3 cells at day 7 and/or at day 14 compared to the control condition. Mature NK cells, which emerge from day 14 in the control cultures, were also greatly reduced in the TXNIP knockdown condition, in which stage 4 and stage 5 NK cells were similarly affected. Overexpression of TXNIP also resulted in reduced cell numbers of HSC, stage 1, and stage 2 cells at day 7 and/or at day 14, but to a lesser extent as compared to the knockdown cultures. Stage 3 and mature NK cell numbers showed no difference compared to the control condition. 

Thus, TXNIP knockdown inhibits the development of human NK cells, whereas overexpression has no effect.

### 2.2. TXNIP Knockdown Affects the Transcriptome of HSC and Stage 3 Cells 

The results of Figure 1 show that TXNIP knockdown in HSC led to decreased NK cell numbers at the end of the differentiation culture, but they also showed that the HSC and stage 1 to 3 cell numbers had already decreased by early time points. This points towards an early effect of TXNIP in the human NK cell developmental process. In order to gain more insight in these effects of TXNIP knockdown, we performed genome-wide transcriptome profiling on two populations at early time points. The first population were HSC sorted from the knockdown and control cultures on day 3 after transduction to detect differentially expressed genes (DEG) in these pluripotent stem cells. As a second population, we opted for day 7 stage 3 cells, as this is the first cell population to appear in the NK cell differentiation process that contains NK cell-committed progenitor cells. 

In the HSC population a total of 372 genes were differentially expressed, with 170 genes being upregulated and 202 being downregulated. Stage 3 cells with decreased TXNIP expression differentially expressed 1847 genes, of which 829 were upregulated and 1018 were downregulated (Figure 2A,B). Comparative analysis of the DEG of the HSC and stage 3 populations showed a great overlap, with 78 and 93 genes that were up- and downregulated, respectively, in both populations, whereas only a total of 5 genes showed opposing differential expression in the two populations (Figure 2C). 

To identify the main biological processes that were affected by TXNIP knockdown, gene ontology (GO) analysis was performed for the up- and downregulated genes in both sequenced populations using Gene Set Enrichment Analysis (GSEA). Cell division or related biological pathways were predominantly present in the top five upregulated genes in both HSC and stage 3 cells. Ribosome biogenesis and cytoplasmic translation were the top listed pathways of downregulated genes of HSC and stage 3 cells, respectively. Respiration pathways completed the five affected biological processes of downregulated genes of both cell populations (Table 1). 

In conclusion, RNA sequencing reveals that knockdown of TXNIP affects the transcriptome of HSC and stage 3 cells.

### 2.3. TXNIP Knockdown Decreases Proliferation, but Has No Impact on Apoptosis

Greatly decreased cell numbers in the TXNIP knockdown cultures of the different developmental stages at day 7 indicated either decreased cell proliferation and/or increased apoptosis. Cell division was the biological process most enriched in the upregulated DEGs of both the HSC and stage 3 cells upon TXNIP knockdown (Table 1 and Figure 3A). Among the upregulated DEGs in the HSC and stage 3 populations were cell cycle inhibitors, including *CDKN1A*, the gene encoding p21, and BTG3, which is a member of the antiproliferative BTG gene family [19,20,21], but also cell cycle activators such as cyclin dependent kinase 6 (*CDK6*) and cyclin E2 (*CCNE2*), which both play a role in G1 phase transition [22,23]. On the other hand, cell division cycle protein 42 (*CDC42*), which favors cell cycle progression [24], cullin 4A (*CUL4A*), which is involved in cell cycle regulation [25] and *PTEN*, a negative regulator of mitotic checkpoint [26], were downregulated. Consistent with the RNA sequencing data, RT-qPCR confirmed higher expression of *CDKN1A* and *BTG3* and lower expression of *CDC42* in both populations (Figure 3B). 

Since the up- and downregulated genes upon TXNIP knockdown contain both positive regulators as well as negative regulators of mitotic cell cycle (Figure 3C), we performed CellTrace experiments to determine the net effect of these TXNIP knockdown DEGs on cell proliferation. After transduction, eGFP^+^ HSC were sorted, labelled with CellTrace Violet, and then cultured in the NK cell differentiation condition. The CellTrace signal was analyzed on day 5 in the early NK cell stages. All early developmental stages in the knockdown condition showed a lower proliferation rate as quantified by the percentage of CellTrace^low^ cells. Overexpression of TXNIP did not significantly affect proliferation (Figure 3D).

To analyze apoptosis, we performed Annexin V plus propidium iodide staining of cells from day 5 knockdown and overexpression cultures. TXNIP knockdown cells did not show a higher apoptosis rate. On the contrary, stage 1 cells had a decreased frequency of early apoptotic cells compared to the control. Additionally, in TXNIP overexpression cultures there was no difference in the frequency of apoptotic cells, except for an increase in late apoptotic stage 2 cells (Figure 3E).

IL-15 plays an important role in the differentiation, proliferation, and survival of NK cells [27]. In mice, TXNIP regulates expression of CD122, the β chain of the IL-2 receptor that also binds IL-15. Indeed, NK cells from *Txnip* knockout mice have reduced CD122 expression, making these cells less responsive to IL-15 compared to wild-type NK cells [17]. However, analysis of the frequency of CD122-expressing cells in the NK cell population of TXNIP knockdown or TXNIP overexpression showed no difference as compared to their respective controls (Figure 3F). When analyzing the CD122 expression level, quantified as the mean fluorescence intensity (MFI), NK cells from day 21 TXNIP knockdown cultures showed lower expression than control NK cells (Figure 3G). 

Taken together, these results indicate that TXNIP knockdown decreases proliferation, while it has no impact on apoptosis.

### 2.4. TXNIP Knockdown Results in Decreased Protein Synthesis

Protein synthesis is closely linked to proliferation. The results of the GO analysis demonstrated that in the HSC population the downregulated DEGs were significantly enriched for the ribosome biogenesis pathway, while in stage 3 cells the cytoplasmic translation pathway was at the top of the enriched processes (Table 1 and Figure 4A). RT-qPCR confirmed decreased expression of *LTV1*, a gene involved in ribosomal small subunit biogenesis [28], and ribosomal RNA processing protein, *RRP15*, in HSC of TXNIP knockdown cultures (Figure 4B). As ribosomal biogenesis is a prerequisite for efficient protein translation, we assessed the de novo protein synthesis rate in TXNIP knockdown cultures. Cells of day 3 and day 7 cultures were treated with O-propargyl-puromycin (OPP), which is incorporated into newly translated proteins, and then OPP was fluorescently labelled to perform flow cytometric analysis. These experiments showed that all developmental stages of TXNIP knockdown cultures that were present, i.e., HSC and stages 1 to 3, displayed a trend of decreased protein synthesis on day 3 of culture, while on day 7 protein synthesis was significantly reduced in all these stages (Figure 4C).

This indicates that TXNIP knockdown reduces the protein synthesis of early developing NK cells. 

### 2.5. TXNIP Knockdown Partially Alters the Phenotype of NK Cells 

Lee et al. showed that the remaining NK cells in *Txnip* knockout mice display greatly reduced expression of NKG2A/C/E and Ly49 receptors [17]. To assess whether TXNIP also influences human NK cell receptor expression, we analyzed the phenotype of NK cells at day 21 of both TXNIP knockdown and overexpression cultures. The expression of NKp30, NKG2A, NKG2C, and NKG2D was modestly elevated in NK cells from the TXNIP knockdown cultures, whereas there was a slight downregulation of KIR (pan-KIR antibody) and NKp44. Other receptors, including the Fc gamma receptor CD16, were unaltered. Overexpression of TXNIP led to a small increased frequency of KIR expression on NK cells, whereas all other examined receptors were similarly expressed as in control cultures (Figure 5A). 

EOMES and TBET are two important transcription factors in NK cell differentiation. During maturation, human NK cells evolve from a TBET^low^EOMES^hi^ to a TBET^hi^EOMES^low^ phenotype [29,30,31]. Comparison of the TBET and EOMES expression profile in the NK cell population of TXNIP knockdown versus control cultures revealed a slight shift to the immature phenotype upon TXNIP knockdown (Figure 5B). However, most other transcription factors known to be important in NK cell differentiation showed equal expression upon knockdown or overexpression of TXNIP compared to their controls (Figure 5C). 

Thus, although TXNIP knockdown influences the expression of some NK cell receptors and of TBET and EOMES, the overall differences in expression of NK receptors and of transcription factors involved in NK cell differentiation are limited.

### 2.6. TXNIP Is Dispensable for NK Cell Functionality

The main functions of NK cells are cytotoxicity and cytokine production. To assess whether NK cells generated in the TXNIP knockdown or overexpression cultures were functionally mature, we evaluated the cytotoxic and cytokine-producing potential of NK cells from day 21 cultures. No significant difference was seen in the cytotoxic capacity of the NK cells from the TXNIP overexpression or knockdown cultures as tested by a chromium release assay using K562 tumor cells as targets (Figure 6A). We also evaluated the degranulation capacity of the NK cells by measuring CD107a expression following coculture with the K562 target cell line (Figure 6B). Although there was a trend of lower CD107a expression in TXNIP knockdown cultures and of higher CD107a expression in TXNIP overexpression cultures, the difference was not significant. Next, the expression levels of perforin and granzyme B in gated NK cells were evaluated. Again, a non-significant trend was observed, with higher perforin and granzyme B expression in the knockdown condition and lower expression in the overexpression condition (Figure 6C). 

To assess cytokine production by NK cells, we stimulated bulk cells from day 21 cultures with either PMA/ionomycin or K562 target cells during 6 h, and with IL-12 plus IL-18 or IL-12, IL-15, and IL-18 for 24 h. Flow cytometric analysis on gated NK cells showed no difference in the percentages of IFN-γ and TNF-α producing cells from knockdown or overexpression cultures (Figure 6D). To analyze the secretion of IFN-γ and TNF-α, NK cells were sorted from day 21 cultures and stimulated using the same cytokine conditions for 24 h, after which the supernatant was harvested and analyzed with ELISA. Both IFN-γ and TNF-α secretion were similar to the control cultures (Figure 6E). 

Thus, TXNIP expression does not affect NK cell effector functions.

## 3. Discussion

In recent years, our understanding of the development of NK cells has rapidly increased with the identification of different developmental stages and of several key factors involved in this process [8]. However, most of our knowledge is derived from genetically modified mice models and, due to interspecies variability, translational research into human NK cells is still urgently required. 

The innate role of NK cells in tumor immunosurveillance made them subject to intensive research in cancer immunotherapy in recent years. The reduced concern for graft-versus-host disease and the possibility of off-the-shelf therapy makes NK cells an appealing candidate for clinical applications [32]. Currently, multiple clinical studies with NK cells for adoptive transfer are ongoing, using peripheral blood NK cells, NK cell lines, or stem cell-derived NK cells [33]. As NK cell-based therapy requires sufficient numbers of functional NK cells, an efficient ex vivo expansion period is essential. Stem-cell derived NK cells offer the ability to manipulate the differentiation strategy [32]. Some methods also include genetic modifications to express chimeric antigen receptors [34]. Increased understanding of human NK cell differentiation will contribute to the improvements of NK cell-based immunotherapies.

Important factors involved in the differentiation of immune cells can often be identified by their high expression during differentiation. In mice, TXNIP is highly expressed in CD122^+^ NK progenitor cells and in mature NK cells [17]. Here, we examined expression of TXNIP during human NK cell development. We show that TXNIP is highly expressed in cord blood-derived HSC, is downregulated in stage 1 and stage 2 cells, and is again upregulated in stage 3 cells. While stage 1 and stage 2 cells are pluripotent, stage 3 cells comprise NK cell-committed progenitors [8]. TXNIP expression remains high in stage 4 and stage 5 cells, which are mature NK cells. This expression pattern is in accordance with that in murine NK cell development and is suggestive for a role of TXNIP in human NK cell development. 

The generation of *Txnip* knockout mice has shown that TXNIP is required for murine NK cell development as these mice have profoundly reduced NK cell numbers in the bone marrow, as well as in the spleen and lung [17]. Here, we show, for the first time, the influence of TXNIP on human NK cell development by stable transduction of human cord blood-derived HSC with TXNIP-specific shRNA to induce knockdown or with TXNIP cDNA to induce overexpression, followed by in vitro NK cell differentiation. TXNIP knockdown and overexpression were confirmed at the RNA and protein level, validating our experimental setup. TXNIP knockdown in human cord blood HSC led to greatly reduced cell numbers of all subpopulations of the HSC progeny, including the multipotent stage 1 and stage 2 cells, the NK cell-committed stage 3 cells, as well as the mature NK cells, i.e., stage 4 and stage 5 cells. While TXNIP overexpression resulted in a minor decrease of stage 1 and stage 2 cells, there was no effect on the stages 3 to 5. This indicates that endogenous TXNIP levels in these cells are sufficient for differentiation and maintenance.

Lee et al. showed that in vitro differentiation of HSC from *Txnip* knockout mice generates mature NK cells that have decreased expression of IL-2Rβ (CD122) as compared to control NK cells [17]. They hypothesize that this reduced CD122 expression at least partially explains the low NK cell numbers in *Txnip* knockout mice. In contrast, we show that the percentage of CD122-positive cells in the mature NK cell population is not different in the TXNIP knockdown cultures as compared to the control cultures, and the CD122 expression level (MFI) is similar at day 14 and partially reduced at day 21 of culture. Together with the fact that the cell numbers of stage 1 and stage 2 cells, which do not yet express CD122, are also greatly reduced in the TXNIP knockdown cultures, this strongly indicates that CD122 expression has no prominent role in TXNIP-dependent human NK cell differentiation.

We performed transcriptome profiling in HSC and stage 3 cells of TXNIP knockdown versus control cultures. Pathway analysis of the RNA sequencing results showed that cell division was the most significantly enriched biological process in the upregulated DEG in both the HSC and stage 3 population. Further analysis of the DEGs showed the presence of both inhibitors as well as activators of cell division. For example, *CDKN1A* and *BTG3* were among the upregulated DEGs, whereas cell division cycle protein 42 (*CDC42*) was downregulated. This was also confirmed with RT-qPCR. CDKN1A is also known as p21, which is a cell cycle inhibitor and downstream effector of p53, while BTG3 is a member of the antiproliferative BTG gene family and has been identified as a direct target of p53 [19,20,21]. While downregulation of BTG3 expression is associated with enhanced cell proliferation, growth, and migration [35], overexpression of BTG3 is associated with suppressed proliferation, reduced cancer invasiveness, and cellular apoptosis in primary cancers and cancer cell lines [36]. CDC42 is a key regulator of the actin cytoskeleton that controls cell motility and polarity and is involved in the regulation of cell cycle progression. Deletion of *CDC42* from Ras-transformed cells decreases cell cycle progression and therefore cell proliferation [24]. Since the affected genes upon TXNIP knockdown contain inhibitors as well as activators of cell division, we performed CellTrace experiments to determine the net effect of these DEGs on cell proliferation. These experiments showed that cell proliferation was lower in all early differentiation stages of the TXNIP knockdown cultures, i.e., stage 1 to stage 3. This was unexpected, as TXNIP is often referred to as a tumor suppressor as its expression is markedly downregulated in various tumors, while overexpression of TXNIP inhibits proliferation of cancer cells [37]. However, recently, a novel pro-erythropoietic role for TXNIP was described, wherein decreased proliferation of erythroblasts was also observed upon TXNIP knockdown [38]. 

A role for TXNIP in apoptosis has also been well established. In the mitochondria, TXNIP competes with proapoptotic protein apoptotic signaling kinase-1 (ASK1) for binding with thioredoxin. Releasing ASK1 from its inhibition by thioredoxin results in the phosphorylation of ASK1 and activation. This, in turn, leads to mitochondrial dysfunction, cytochrome-c release and cleavage of caspase-3, and apoptosis [14,39]. While the effects of TXNIP knockdown or overexpression in our experiments on apoptosis were minimal, they concur the established role of TXNIP in apoptosis, but they probably are not linked to the drastically reduced cell numbers of all NK cell developmental stages that we observed. 

Transcriptome analysis upon knockdown of TXNIP also revealed significant downregulation of genes related to ribosome biogenesis in HSC and to protein translation in stage 3 cells. Consistent with the results obtained by RNA-seq, we observed a trend in lower protein synthesis in the early progenitor stages on day 3 of the TXNIP knockdown cultures and a significantly decreased protein synthesis rate on day 7. Ribosome biogenesis and protein translation are tightly coordinated and are essential for cell growth, proliferation, differentiation, and development. Impairment of these cellular processes causes cells to shut down their cell cycle to avoid incomplete growth and unprepared division [40]. Our hypothesis is that the reduced protein synthesis rate causes, at least in part, the lower proliferation upon TXNIP knockdown.

Our results show that TXNIP has limited effects on the terminal differentiation of human NK cells as the expression of only a few NK receptors was modestly affected, whereas other receptors, including CD16, were unaffected. Similarly, analysis of expression of a large panel of transcription factors known to have a role in human NK cell differentiation showed that only expression of TBET and EOMES was marginally altered in NK cells from TXNIP knockdown cultures. *Txnip*-deficient mice exhibit strongly reduced expression of all Ly49 receptors [17] and the final NK cell maturation stage (CD27^low^CD11b^+^) tended to be increased [41]. Ly49 receptors are the murine functional homologs of human KIR. We only observed a small decrease of KIR expression on the NK cells of TXNIP knockdown cultures. KIR are mainly expressed on CD16-positive NK cells, and the latter population was unaltered upon TXNIP knockdown. One restriction of the in vitro NK cell differentiation culture model is that, although KIR- and CD16-expressing NK cells are eventually obtained, the percentage of these cells is rather limited compared to that of PBMC NK cells. We thus cannot exclude that TXNIP does have a role in terminal NK cell differentiation in vivo. However, our functional experiments show that NK cells from the TXNIP knockdown versus control cultures display no difference in cytotoxic capacity against tumor cells nor cytokine production. At first sight, this might be in contrast to the murine situation, as it was originally reported that NK cells from *Txnip* knockout mice display low cytotoxicity against tumor cells [17]. However, this original finding was achieved comparing the cytotoxicity of total splenocytes of *Txnip* knockout versus wild type mice, and not taking into account the lower NK cell percentage in the splenic knockout cells. More recently, published research of the same group showed no difference in cytotoxicity when using purified NK cells from *Txnip* knockout mice [41]. This is thus in agreement with our findings in humans. Taken together, this indicates that TXNIP, both in mice and humans, has no effect on functional maturation of NK cells. 

In conclusion, our results show that TXNIP promotes human NK cell differentiation. It is required for protein synthesis and thereby probably affects proliferation of the early NK cell differentiation stages. In contrast, its effects on terminal NK cell maturation are minimal, and TXNIP is dispensable for NK cell function.

## 4. Materials and Methods

### 4.1. Viral Constructs

To knockdown the expression of TXNIP, a TXNIP-specific encoding shRNA (5′-CCAACTCAAGAGACAAAGAAA-3′) containing vector with a pLKO.1 backbone (Mission shRNA; Sigma Aldrich, St. Louis, MO, USA) was used. This lentiviral vector contained a puromycin resistance gene that was replaced by the enhanced green fluorescent protein (eGFP) reporter gene. After validation of the construct, viral supernatant was collected 48 h and 72 h after transfecting the lentiviral shRNA vectors together with pCMV-VSV-G envelope and p8.91 packaging vectors in HEK293T cells using JetPEI (Polyplus transfection, Illkirch, France). A non-targeting shRNA sequence was used as control. 

To induce overexpression, TXNIP-encoding gBlocks (Integrated DNA Technologies, Newark, NJ, USA) were cloned in the pCR-blunt vector using the Zero Blunt PCR Cloning kit (Thermo Fisher Scientific, Waltham, MA, USA), followed by subcloning into the LZRS-IRES-eGFP vector [42]. After validation of the construct by sequencing, a viral supernatant was collected 2, 6, and 14 days after transfecting the retroviral vectors in Phoenix A cells using calcium phosphate transfection. The empty LZRS-IRES-eGFP vector was used as control.

### 4.2. Isolation of Hematopoietic Stem Cells

CD34^+^ cells were isolated from human umbilical cord blood (Cord Blood Bank, University Hospital Ghent, Ghent, Belgium). Cord blood was obtained with informed consent in accordance with the Declaration of Helsinki, and usage was approved by the Ethics Committee of the Faculty of Medicine and Health Sciences (Ghent University, Ghent, Belgium). After isolation of mononuclear cells by Lymphoprep (Stem Cell Technologies, Grenoble, France) density gradient centrifugation, CD34^+^ cells were purified using Magnetic Activated Cell Sorting (MACS; Miltenyi Biotec, Leiden, The Netherlands). After 48 h of preculture in Iscove’s Modified Dulbecco’s Medium (IMDM; Thermo Fisher Scientific) containing fetal calf serum (FCS; Biowest, Nuaillé, France) (10%), penicillin (100 U/mL), streptomycin (100 µg/mL), and glutamine (2 mM) (all from Life Technologies, Grand Island, NY, USA), supplemented with thrombopoietin (TPO) (20 ng/mL), stem cell factor (SCF; Peprotech, London, UK) (100 ng/mL), and FMS-like tyrosine kinase 3 ligand (Ftl3L; R&D Systems, Minneapolis, MN, USA) (100 ng/mL), the cells were transferred to RetroNectin (2 µg/cm^2^) (Takara Bio, Saint-Germain-en-Laye, France) coated plates and viral supernatant was added, followed by spinoculation at 950 g during 90 min at 32 °C. In case of lentiviral transduction, polybrene (Sigma Aldrich) (8 µg/mL) was added during the transduction. Twenty-four hours after lentiviral transduction, the medium was refreshed to remove polybrene. eGFP^+^ hematopoietic stem cells (HSCs), defined as CD34^+^lineage^−^(CD3/CD14/CD19/CD56)CD45RA^−^ cells, were sorted to high purity 48 h after transduction using a FACS ARIA II cell sorter (BD Biosciences, San Jose, CA, USA).

### 4.3. Coculture Systems

Following FACS sorting, eGFP^+^ HSCs were plated on the mitomycin C-inactivated murine embryonic liver cell line EL08-1D2, which was kindly provided by E. Dzierzak (Erasmus University MC, Rotterdam, The Netherlands). Cells were co-cultured in NK cell coculture medium containing Dulbecco’s Modified Eagle Medium (DMEM) and Ham’s F-12 nutrient mixture (2:1 ratio) (all from Thermo Fisher Scientific), supplemented with penicillin (100 U/mL), streptomycin (100 µg/mL), glutamine (2 mM), sodium pyruvate (10 mM) (Thermo Fisher Scientific), heat-inactivated human AB serum (20%) (Biowest), β-mercaptoethanol (24 µM), ascorbic acid (20 µg/mL), ethanolamine (50 µM), and sodium selenite (50 ng/mL) (all from Sigma Aldrich). The cytokines IL-3 (5 ng/mL) (R&D systems), IL-7 (20 ng/mL), IL-15 (10 ng/mL), SCF (20 ng/mL), and Ftl3L (10 ng/mL) were added to the culture medium. On day 7 of culture, the medium was refreshed by addition of equal volumes of fresh medium supplemented with cytokines (except IL-3). On day 14 of culture, the cells were split and transferred to new inactivated EL08-1D2 stromal cells. Cultures were maintained in a humidified atmosphere of 5% CO_2_ at 37 °C. 

EL08-1D2 cells were maintained on 0.1% gelatin-coated plates at 32 °C in Myelocult M5300 medium (50%) (Stem Cell Technologies), α-MEM (35%), heat-inactivated FCS (15%), supplemented with penicillin (100 U/mL), streptomycin (100 µg/mL), glutamine (2 mM), and β-mercaptoethanol (10 μM). Cell proliferation was blocked by the addition of mitomycin C (10 μg/mL) to the culture medium for 2–3 h, followed by a thoroughly rinsing of the cells before harvesting using trypsin-EDTA (Lonza, Bazel, Switzerland). Cells were plated at a density of 50,000 cells per well of a 0.1% gelatin-coated tissue culture-treated 24-well plate at least 24 h before adding HSCs or differentiating NK cells.

### 4.4. Flow Cytometry Analysis and Sorting

Cells were harvested by forceful pipetting at indicated timepoints and immunostained for phenotypical analysis. In vitro NK developmental subsets were identified and analyzed using the following gating strategy on eGFP^+^ cells: HSC (CD34^+^CD45RA^−^), stage 1 (CD34^+^CD45RA^+^CD117^−^), stage 2 (CD34^+^CD45RA^+^CD117^+^), stage 3 (CD34^−^CD94^−^CD117^+^HLA-DR^−^NKp44^−^), stage 4 (CD11a^+^CD56^+^CD94^+^CD16^−^), and stage 5 (CD11a^+^CD56^+^CD94^+^CD16^+^). 

To stain intracellular and intranuclear proteins, the BD Cytofix/Cytoperm (BD Bioscience) and Foxp3/Transcription Factor Staining Buffer Set (Thermo Fisher Scientific) were used, respectively.

Before staining, the cells were blocked with anti-mouse FcRgII/III (clone 2.4.G2) and human IgG (Miltenyi Biotec). To discriminate living and dead cells in cell membrane and intracellular or -nuclear staining, propidium iodide and Fixable Viability Dye eFluor™ 566 (Thermo Fisher Scientific) were used, respectively. 

For apoptosis assays, cells were washed in annexin binding buffer and stained with annexin V-APC (Thermo Fisher Scientific). 

Cells were analyzed on an LSRII flow cytometer (BD Biosciences). For sorting, a FACSARIA was used. FlowJo_v10.8.1 (Ashland, OR, USA) was used for analysis. Utilized antibodies are listed in Appendix A.

### 4.5. Cell Proliferation Assay

Cell proliferation was determined using the CellTrace™Violet Cell Proliferation kit (Thermo Fisher Scientific) following the manufacturer’s instructions and analyzed by flow cytometry at the indicated time point.

### 4.6. Protein Synthesis Assay 

Protein synthesis rate was measured using the Click-iT Plus OPP Alexa Fluor 647 Protein Synthesis Assay (Thermo Fisher Scientific) according to the manufacturer’s protocol. Briefly, O-propargyl puromycin (OPP; 5 µM) was added to the cells for 30 min at 37 °C, 5% CO_2_. After incubation, the cells were harvested and washed with PBS and stained extracellularly. Cells were then fixed and permeabilized using the Cytofix/Cytoperm Fixation Permeabilization Kit (BD Biosciences). Next, the cells were incubated for 30 min at room temperature in the dark with a 100 µL Click-iT reaction cocktail, prepared as instructed by the manufacturer. After washing, cells were analyzed with flow cytometry on a FACSSymphony (BD Biosciences). As a control, protein synthesis was blocked by treatment with cycloheximide (100 µg/mL) for 30 min before OPP was added. Geometric mean fluorescence intensity was used as an indicator of the relative translation.

### 4.7. Cytokine Production and Secretion

For flow cytometric analysis of cytokine production, coculture cells of day 21 were stimulated in bulk during 6 h with phorbol myristate acetate (PMA; 5 ng/mL) and ionomycin (1 µg/mL) or with K562 cells at an effector to target ratio (E:T) of 1:1, or during 24 h with IL-12 plus IL-18 (both 10 ng/mL) or IL-12, IL-18, and IL-15 (4 ng/mL). In the last 4 h of incubation, brefeldin A (BD Golgiplug, BD Biosciences) was added. After harvesting, cells were stained for NK surface markers and subsequently fixed and permeabilized for intracellular staining of IFN-γ and TNF-α. For analysis of cytokine secretion, sorted mature eGFP^+^ NK cells (CD45^+^CD56^+^CD94^+^) from day 21 cultures were stimulated with IL-12 plus IL-18 or IL-12, IL-18, and IL-15 (same concentrations as indicated above). After 24 h, the supernatant was collected and analyzed for cytokine secretion with IFN-γ ELISA assay (PeliKine-Tool Set, Sanquin, Amsterdam, The Netherlands) and TNF-α ELISA assay kits (TMB ELISA Development Kit, Peprotech).

### 4.8. Cytotoxicity Assay

K562 target cells (10^6^) were labelled with 100 µCi of Na_2_^51^CrO_4_ (Perkin Elmer, Waltham, MA, USA) for 1 h at 37 °C, 5% CO_2_. Labelled cells were washed three times in medium and resuspended in NK cell coculture medium medium. Target cells were co-incubated with sorted eGFP^+^ NK cells at E:T ratios of 3, 1, 0.3, 0.1, and 0.03. Spontaneous release was measured by incubating target cells with medium alone, while maximum release was measured by incubating target cells in 1% Triton X-100. After 4 h, the supernatant was harvested and mixed with scintillation fluid (Perkin Elmer). Radioactivity was measured with a 1450 LSC&Luminescence Counter (Wallac Microbeta Trilux, Perkin Elmer). The mean percentage of cytotoxic activity of triplicates was calculated.

### 4.9. Degranulation Assay

K562 target cells were co-incubated with bulk cells from day 21 of coculture at an E:T ratio of 1:1. After 2 h, cells were harvested and subsequently stained for CD56, CD94, and CD107a. Cell membrane CD107a expression on gated NK cells was analyzed by flow cytometry.

### 4.10. Western Blot

Sorted cells (Appendix A) were lysed in RIPA buffer and protein concentration was determined using the DC protein assay (Bio-RAD, Hercules, CA, USA). Denatured protein was loaded on a Bolt 4–12% Bis-Tris Plus gel (Thermo Fisher Scientific) and transferred to a PVDF membrane (Thermo Fisher Scientific). After blocking, the membrane was incubated with the primary antibody at 4 °C overnight, followed by incubation with the secondary antibody for 1 h. For visualization, anti-mouse conjugated horseradish peroxidase secondary antibody (#NA931, Sigma Aldrich) was used. Protein level quantification was performed using ImageJ software (National Institutes of Health). The primary antibodies used were anti-TXNIP (#K0204-3, Medical and biological laboratories, Woburn, MA, USA, dilution 1:500) and anti-VINCULIN (#V9131, Sigma Aldrich; dilution 1:10,000)

### 4.11. qPCR Analysis

Total RNA from sorted cells was extracted using the RNeasy Micro kit (Qiagen, Hilden, Germany) and converted into cDNA using the iScript™ Advanced cDNA synthesis Kit (Bio-RAD). Quantitative PCR was performed using the LightCycler 480 SYBR Green I Master mix (Roche, Bazel, Switzerland) on a LightCycler 480 real-time PCR system (Roche). The housekeeping genes GAPDH and either TBP or YHWAZ were used as normalization genes to calculate gene expression levels. Utilized primers are listed in Appendix A.

### 4.12. Library Preparation, RNA Sequencing, and Analysis

For transcriptome analysis, day 3 HSC (eGFP^+^CD34^+^lineage^−^CD45RA^−^) and day 7 stage 3 cells (eGFP^+^CD45^+^CD34^−^CD117^+^CD94^−^NKp44^−^HLA-DR^−^) were sorted (Appendix A) and RNA was isolated using the RNeasy Micro kit (Qiagen). The concentration and quality of the extracted RNA was checked using the ‘Quant-it ribogreen RNA assay’ (Life Technologies) and the RNA 6000 nano chip (Agilent Technologies, Santa Clara, CA, USA), respectively. The RNA sequencing libraries of five biological replicates of the HSC and stage 3 cells were prepared using the QuantSeq 3′ mRNA-Seq Library Prep Kit (Lexogen, Vienna, Austria) using 25 ng and 20.5 ng of input RNA, respectively. Libraries were quantified by qPCR, according to Illumina’s protocol ‘Sequencing Library qPCR Quantification protocol guide’, version February 2011. A High Sensitivity DNA chip (Agilent Technologies) was used to control the library’s size distribution and quality. Sequencing was performed on a high throughput Illumina NextSeq 500 flow cell, generating 75 bp single reads. Per sample, on average, 4.2 × 10^6^ ± 1.1 × 10^6^ and 3.7 × 10^6^ ± 0.8 × 10^6^ reads were generated for the HSC and stage 3 population, respectively. Quality control of these reads was performed with FastQC [43]. Fastq files were aligned to human reference genome GRCh38 using STARv2.42 and gencode v35 as guide gtf. Counts were generated on the fly by STAR. Differential expression analysis was performed using Deseq2 with Wald test for *p*-value calculation [44]. Genes with padj < 0.05 were considered significantly differential. GSEA was performed using the GSEA software tool v4.2.1 of the Broad Institute [45,46]. The ‘GSEAPreranked’ module was run using standard parameters and 1000 permutations.

### 4.13. Statistical Analysis and Software

Data were plotted and statistical analyses were performed using GraphPad Prism v8.3.1 software (GraphPad Software, San Diego, CA, USA). Results were considered statistically significant when *p* < 0.05. All error bars represent the standard error of the mean (SEM).

## Figures and Tables

**Figure 1 ijms-23-11345-f001:**
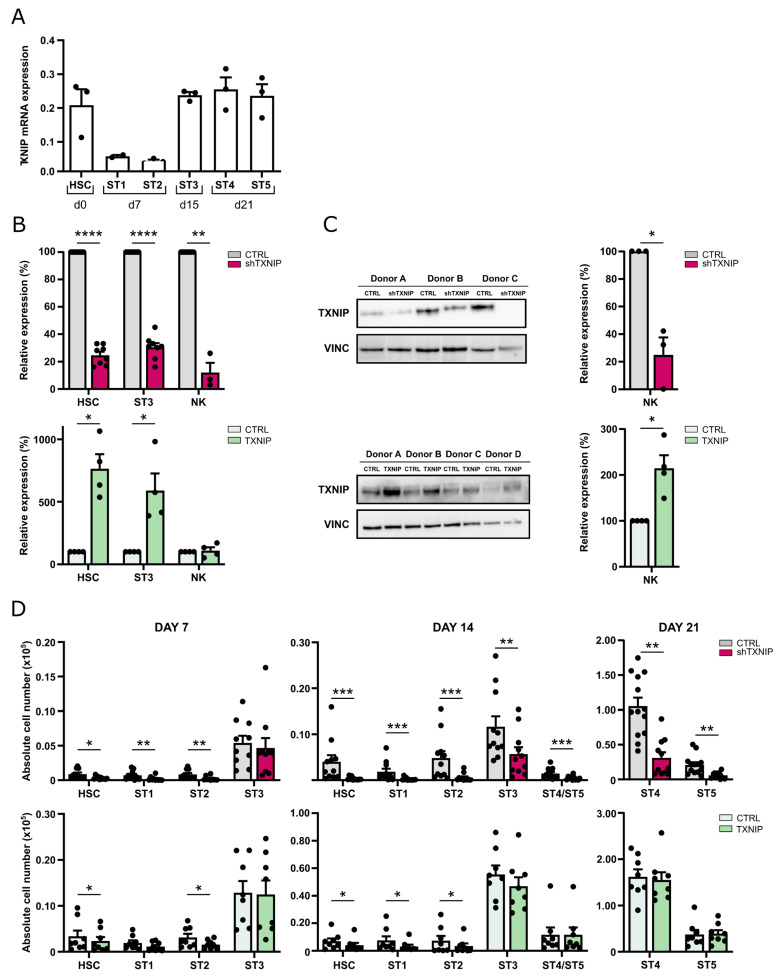
TXNIP promotes human NK cell development. (**A**) TXNIP expression was determined using RT-qPCR in the indicated successive NK cell developmental stages sorted from HSC-based in vitro NK cell differentiation cultures: HSC (Lin^−^CD34^+^CD45RA^−^) sorted on day (d)0, stage (ST) 1 (CD34^+^CD45RA^+^CD117^−^) and ST2 (CD34^+^CD45RA^+^CD117^+^) on d7, ST3 (CD34^−^CD117^+^CD94^−^HLA-DR^−^NKp44^−^) on d15, and ST4 (CD56^+^CD94^+^CD16^−^) and ST5 (CD56^+^CD94^+^CD16^+^) on d21 (mean ± SEM; n = 2–3). (**B**) Upon transduction with TXNIP shRNA (top) or TXNIP overexpression vectors (bottom), sorted eGFP^+^ HSC were cultured in vitro in NK cell-specific culture conditions. Relative TXNIP expression was determined using RT-qPCR in sorted HSC at d3, in ST3 NK cells at d7, and in NK cells at d21. Expression is reported as mean percentage relative to the control condition (set at 100%) (mean ± SEM; n = 3–5). (**C**) TXNIP protein expression was determined by Western blot analysis in sorted NK cells from d21 knockdown (top) or overexpression (bottom) cultures. Scatter plot (right) shows quantification of the TXNIP protein levels normalized to vinculin (VINC) and reported relative to the control condition (set at 100%) (mean ± SEM; n = 3–4). (**D**) Upon transduction with TXNIP shRNA (top), TXNIP overexpression vector (bottom) or their appropriate controls, eGFP^+^ HSC, were sorted and in vitro cultured in NK cell-specific culture conditions. At the indicated time points, absolute cell numbers of HSC and stages 1 to 5 were determined (mean ± SEM; n = 8–11). *, **, *** and **** represent statistical significance compared to control transduced cultures with *p* < 0.05, *p* < 0.01, *p* < 0.001 and *p* < 0.0001, respectively.

**Figure 2 ijms-23-11345-f002:**
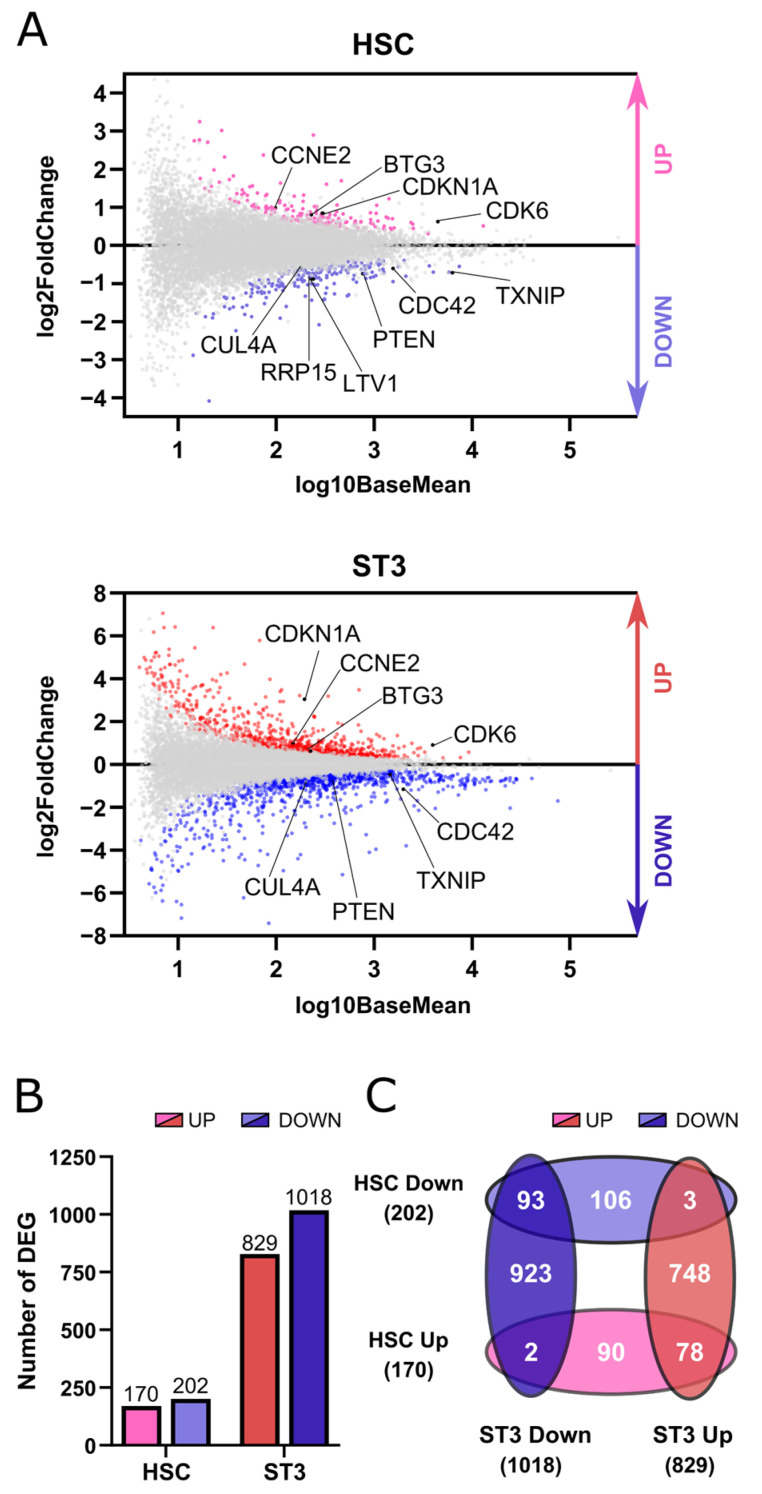
TXNIP knockdown affects the transcriptome of HSC and stage 3 cells. RNA sequencing was performed on five biological replicates of HSC and ST3 cells sorted from TXNIP knockdown cultures on d3 and d7, respectively. (**A**) MA plots showing up- (red) and downregulated (blue) genes in HSC and ST3 cells. (**B**) Bar graphs showing the number of differentially expressed genes (DEG) in HSC and ST3 that were up- (red) or downregulated (blue). (**C**) Venn diagram showing the overlap between the DEGs in the HSC and ST3 populations.

**Figure 3 ijms-23-11345-f003:**
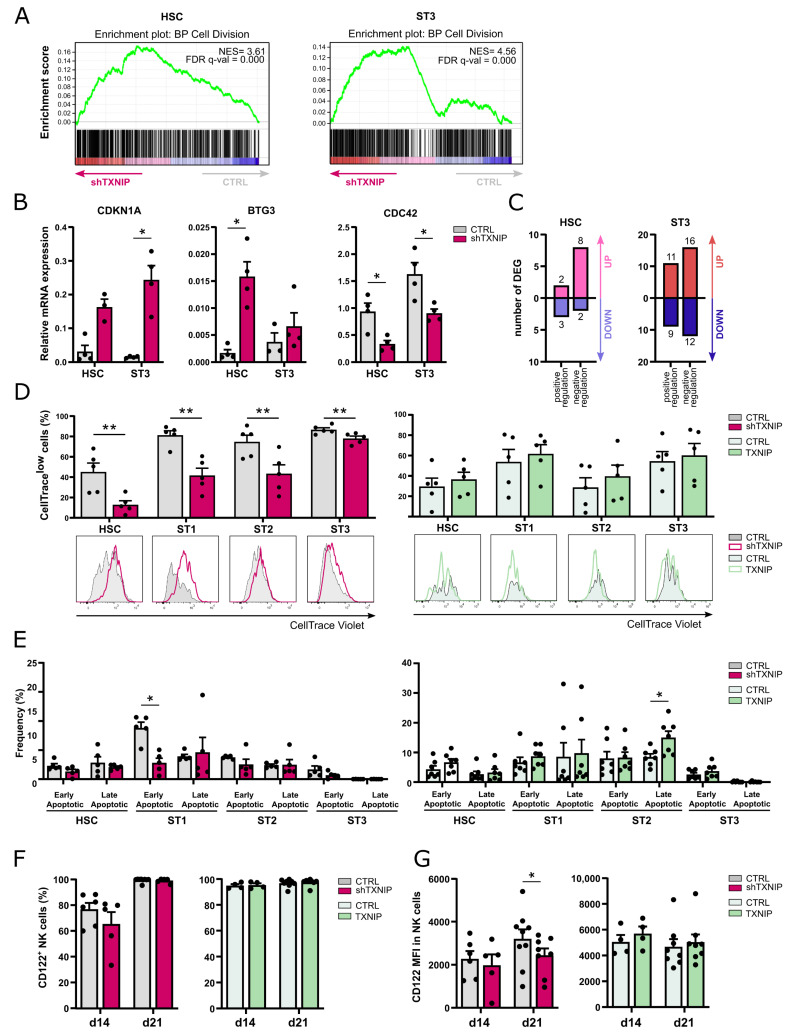
Decreased proliferation of early NK cell developmental stages upon TXNIP knockdown. (**A**) Gene set enrichment analysis plots of the Gene Ontology cell division pathway upon TXNIP knockdown in HSC on d3 (left) and ST3 cells on d7 (right). NES: normalized enrichment score. (**B**) The relative expression of *CDKN1A*, *BTG3*, and *CDC42* in d3 HSC and d7 ST3 was determined using RT-qPCR (mean ± SEM; n = 3–4). (**C**) The number of up- and downregulated DEGs in the Gene Ontology positive and negative regulation of mitotic cell cycle pathways in HSC on d3 (left) and ST3 cells on d7 (right). (**D**,**E**) HSC were transduced with TXNIP knockdown or overexpression vectors. (**D**) eGFP^+^ HSC sorted after transduction (d0), were labelled with CellTrace Violet and cultured in NK cell-specific conditions. On day 5, the CellTrace Violet signal was assessed in gated HSC and ST1 to ST3 of knockdown (left) and overexpression (right) cultures with flow cytometry. The frequency of CellTrace^low^ cells is indicated (mean ± SEM; n = 5). Overlaid histograms of representative samples are shown. (**E**) Apoptosis was assessed on day 5 of culture by flow cytometry in the indicated developmental stages of knockdown (left) and overexpression (right) cultures by staining with propidium iodide (PI) and Annexin V. The percentage of early (Annexin V^+^PI^−^) and late apoptotic cells (Annexin V^+^PI^+^) is shown (mean ± SEM; n = 5). (**F**) The frequency and (**G**) the mean fluorescence intensity (MFI) of CD122 expression in ST4 + 5 NK cells were determined by flow cytometry on d14 and d21 of TXNIP knockdown (left) and overexpression cultures (right) (mean ± SEM; n = 4–8). * and ** represent statistical significance compared to control transduced conditions with *p* < 0.05 and *p* < 0.01, respectively.

**Figure 4 ijms-23-11345-f004:**
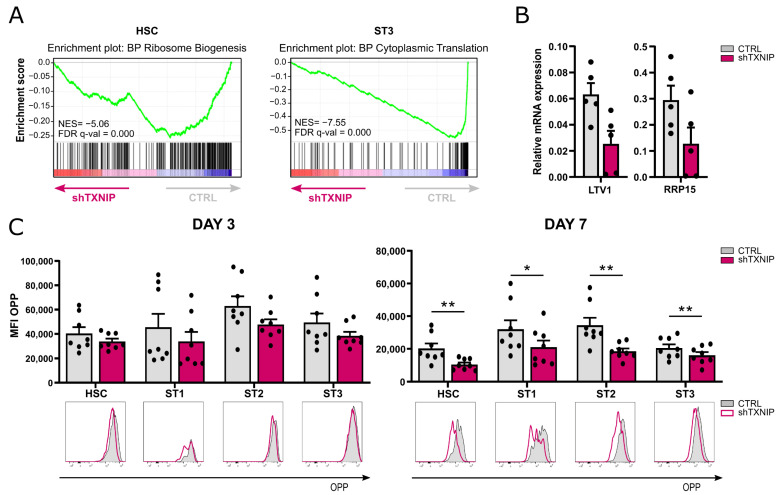
Decreased protein synthesis upon TXNIP knockdown. (**A**) Gene set enrichment analysis plots of Gene Ontology ribosome biogenesis pathway in HSC (left) and of cytoplasmic translation in ST3 cells (right) upon TXNIP knockdown. NES: normalized enrichment score. (**B**) The relative expression of *LTV1* and *RRP15* in d3 HSC was determined using RT-qPCR (mean ± SEM; n = 4–5). (**C**) The protein synthesis rate of cells from TXNIP knockdown versus control cultures was analyzed using the O-propargyl puromycin (OPP) kit. On d3 (left) and d7 (right), cells were incubated with OPP for 30 min, followed by a Click-iT reaction with an Alexa Fluor 647 fluorophore and flow cytometric quantification. The OPP mean fluorescence intensity (MFI) in the indicated developmental stages is shown (mean ± SEM; n = 8). Overlaid histograms of OPP fluorescence of representative samples are displayed. * and ** represent statistical significance compared to control transduced conditions with *p* < 0.05 and *p* < 0.01, respectively.

**Figure 5 ijms-23-11345-f005:**
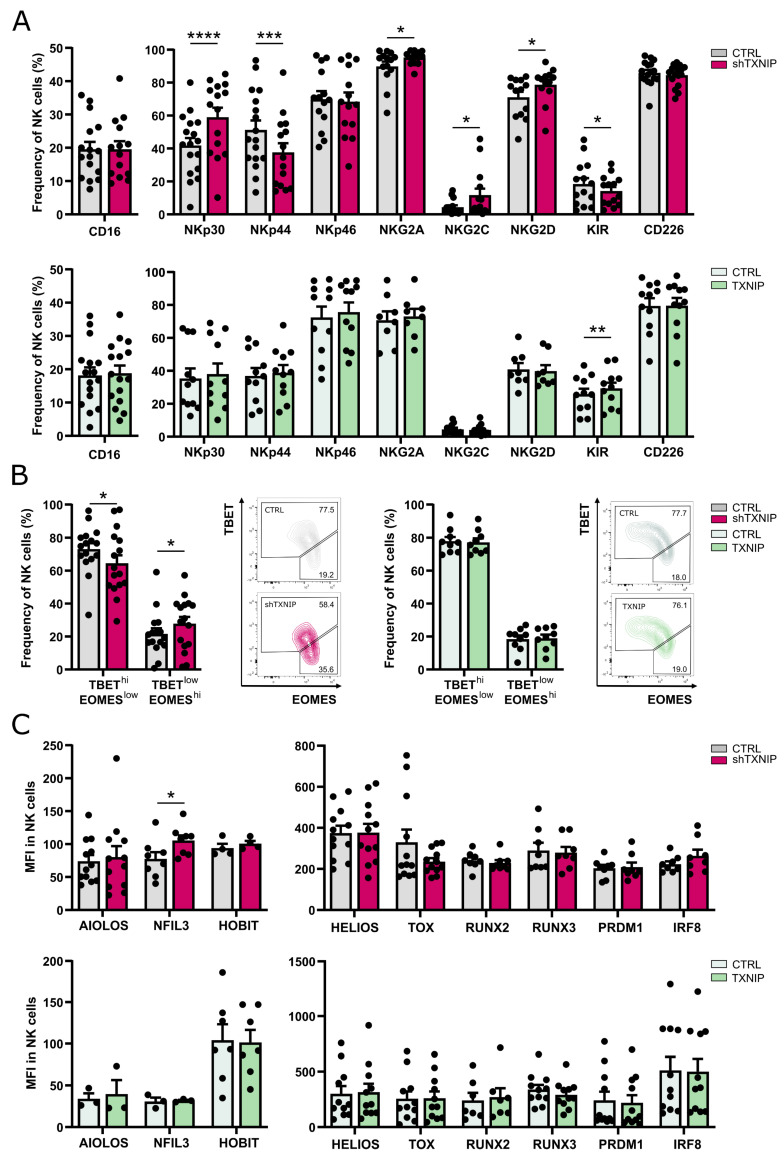
Partially altered phenotype of NK cells upon TXNIP knockdown. (**A**) On day 21 of culture, NK cells (eGFP^+^CD45^+^CD11a^+^CD56^+^CD94^+^) of TXNIP knockdown (top) or overexpression (bottom) conditions were analyzed for expression of the indicated cell membrane NK cell markers by flow cytometry. The frequency of expression is shown (mean ± SEM; n = 8–17). (**B**) Percentage of NK cells with a TBET^hi^EOMES^low^ or TBET^low^EOMES^hi^ phenotype on day 21 of TXNIP knockdown (left) and overexpression (right) cultures (mean ± SEM; n = 9–16). Representative dot plots are shown. (**C**) The mean fluorescence intensity (MFI) of the indicated transcription factors was determined by flow cytometry on day 21 in NK cells of TXNIP knockdown (top) and overexpression (bottom) cultures (mean ± SEM; n = 4–12). *, **, *** and **** represent statistical significance compared to control transduced cultures with *p* < 0.05, *p* < 0.01, *p* < 0.001 and *p* < 0.0001, respectively.

**Figure 6 ijms-23-11345-f006:**
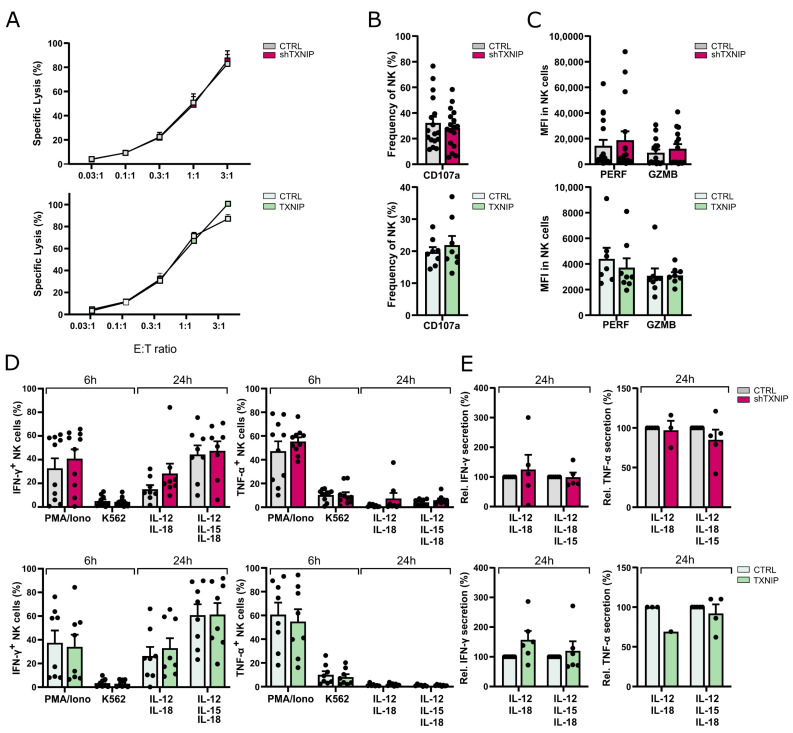
Dispensable role for TXNIP in NK cell functionality. The functionality of NK cells of TXNIP knockdown and overexpression conditions was examined on day 21 of culture. (**A**) NK cell cytotoxicity was assessed using a 51-chromium release assay. NK cells (eGFP^+^CD45^+^CD56^+^CD94^+^) were sorted and incubated for 4 h with K562 target cells at the indicated effector to target (E:T) ratio. The percentage of specific target cell lysis is shown (mean ± SEM; n = 3–11). (**B**) NK cell degranulation was measured by flow cytometric analysis of CD107a cell membrane expression in NK cells (eGFP^+^CD45^+^CD56^+^CD94^+^) after 2 h incubation with K562 target cells at a 1:1 E:T ratio (mean ± SEM; n = 8–17). (**C**) Expression (MFI) of the cytotoxic mediators perforin (PERF) and granzyme B (GZMB) in NK cells (CD45^+^CD56^+^CD94^+^) (mean ± SEM; n = 8–16). (**D**) IFN-γ and TNF-α production was analyzed with flow cytometry in gated NK cells after stimulation of bulk cells with PMA/Ionomycin or by coculture with K562 cells for 6 h, or after 24 h stimulation with either IL-12 plus IL-18 or IL-12, IL-15, and IL-18. Brefeldin A was added during the last 4 h of stimulation (mean ± SEM; n = 7–10). (**E**) NK cells (eGFP^+^CD45^+^CD56^+^CD94^+^) were sorted and stimulated with IL-12 plus IL-18 or IL-12, IL-15, and IL-18. After 24 h, the supernatant was harvested, and IFN-γ and TNF-α secretion were analyzed by ELISA. Cytokine secretion is reported as mean percentage relative to the control condition (set at 100%) (mean ± SEM; n = 1–6).

**Table 1 ijms-23-11345-t001:** The top five enriched biological processes in up- and downregulated differentially expressed genes.

	GO ID	Term	NES	FDR q-val
**HSC**				
**Upregulated**				
	GO:0051301	Cell Division	3.61	0.0000
	GO:0042060	Wound Healing	3.21	0.0000
	GO:0016050	Vesicle Organization	3.08	0.0019
	GO:0000226	Microtubule Cytoskeleton Organization	2.92	0.0029
	GO:0050000	Chromosome Localization	2.90	0.0025
**Downregulated**				
	GO:0042254	Ribosome Biogenesis	−5.06	0.0000
	GO:0034660	ncRNA Metabolic Process	−4.84	0.0000
	GO:0009060	Aerobic Respiration	−4.72	0.0000
	GO:0022613	Ribonucleoprotein Complex Biogenesis	−4.64	0.0000
	GO:0045333	Cellular Respiration	−4.47	0.0000
**Stage 3**				
**Upregulated**				
	GO:0051301	Cell Division	4.56	0.0000
	GO:0000226	Microtubule Cytoskeleton Organization	3.83	0.0000
	GO:0007346	Regulation Of Mitotic Cell Cycle	3.58	0.0000
	GO:0044770	Cell Cycle Phase Transition	3.32	0.0000
	GO:0044772	Mitotic Cell Cycle Phase Transition	3.32	0.0000
**Downregulated**				
	GO:0002181	Cytoplasmic Translation	−7.55	0.0000
	GO:0006119	Oxidative Phosphorylation	−4.18	0.0000
	GO:0009060	Aerobic Respiration	−4.10	0.0000
	GO:0042773	ATP Synthesis Coupled Electron Transport	−3.95	0.0000
	GO:0022900	Electron Transport Chain	−3.87	0.0000

NES: normalized enrichment score.

## Data Availability

RNA-seq data are accessible on GEO with accession number GSE212024. For original data, please contact georges.leclercq@ugent.be.

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
