# Peer review of "TXNIP Promotes Human NK Cell Development but Is Dispensable for NK Cell Functionality"

_ijms, 2022, doi:10.3390/ijms231911345_

Round 1

Reviewer 1 Report

Authors of the manuscript investigated the role TXNIP in the differentiation and functional maturation of human NK cells. The experimental design was appropriate and straightforward. The manuscript is relatively short but well-written. Knockdown and overexpressing cells for TXNIP were generated and used for entire experiments. A couple of comments were added, which, in my opinion need to be addressed to improve the manuscript. 

1.     As shown in Fig1. A. TXNIP gene expression level was determined at each indicated time points using the markers described. Differentiation stages have been defined based on these grouping. Cell numbers were determined for each stage and for all three time points. Is there any reason why cells of ST3+day7 instead of ST3+day15 or ST2+day7 were sorted for other experiments, i.e. RNA seq? One short paragraph, explaining rationale for selected stage and time point would be helpful to understand the results better, especially in the context of differentiation procedure.  

2.     It would be interesting to discuss about more details of upregulated genes of cell division and/or genes related to apoptosis detected shTXNIP cells from RNA seq analysis. Results seem to indicate that both activator and inhibitor of cell cycle progression play together. Overall presentation of data is very descriptive. Results could be more clearly concluded at the end of each section. Many parts of discussion section could be moved to the results part. Short discussion about NK cell-mediated immunotherapy would improve the discussion part. 

Reviewer 2 Report

In the manuscript the authors describes the effects of TXNIP knockdown on NK cells. The topic is of high interest and the manuscript is well presented however, few points should be addressed before considering it for publication:

1) It would be essential to show the FACS analyses of all sorted populations

Round 2

Reviewer 2 Report

The authors fully replied to the comments. I suggest the manuecript for publication

Author Response

Thanks